# The female side of pharmacotherapy for ADHD—A systematic literature review

**Francien M. Kok** [1]*, **Yvonne Groen** [1], **Anselm B. M. Fuermaier** [1], **Oliver Tucha** [1,2]

**1** Department of Clinical and Developmental Neuropsychology, University of Groningen, Groningen, The Netherlands, **2** Department of Psychiatry and Psychotherapy, University Medical Center Rostock, Rostock, Germany

* f.m.kok@rug.nl

**Data Availability Statement:** All relevant data are within the manuscript and its Supporting Information files.

**Funding:** The author(s) received no specific funding for this work.

## Abstract

### Objective

This comprehensive review examined sex differences in prescription rates and efficacy or effectiveness of pharmacotherapy treatment in girls and women with attention deficit hyperactivity disorder (ADHD), while identifying gaps in the scientific knowledge on this topic.

### Method

A rigorous electronic database search was carried out in order to identify all published studies on female-specific effects of stimulants and non-stimulants in the treatment of ADHD. In total, 2672 studies were screened of which 21 studies (seven on prescription rates, 14 on effects of pharmacotherapy) met the inclusion criteria and were included for analysis.

### Results

In all seven studies on ADHD prescription rates, girls received significantly less prescriptions than boys, a difference however no longer seen in adults with the exception of one study. Each of the 14 studies on effectiveness / efficacy found at least one sex-difference in the effects of ADHD pharmacotherapy.

### Conclusion

Several sex-differences are demonstrated in the prescription, usage and efficacy /effectiveness of both stimulant and non-stimulant ADHD pharmacotherapy. A single daily use of MPH may possibly not be optimal for girls with ADHD and ATX may be a promising medication for girls and women with ADHD. The robustness of this result requires further investigation.

## Introduction

Attention deficit hyperactivity disorder (ADHD) is a neurodevelopmental disorder that often persists into adulthood and involves a persistent pattern of inattention and/or hyperactivity-impulsivity that interferes with functioning or development [1]. The impact of ADHD

**Competing interests:** The authors have declared that no competing interests exist.

symptoms can interfere with development and functioning in several domains, and this impact has been found to differ across the sexes [2–8]. The worldwide prevalence estimates of ADHD in school-age children and adolescents vary considerably across countries, ranging from very low rates of 0.6% in girls and 2.8% in boys [9] to high rates of 10.5% in girls and 14.4% in boys [10]. Female-to-male ratios reported also vary considerably with female-to-male ratios between 1:3 and 1:1.5 in population-based studies [11–13] and between 1:5 and 1:9 in clinical samples [14]. ADHD has been diagnosed more frequently across both sexes in the last decade and a half [15, 16].

Females with ADHD were underdiagnosed for many years, partly due to their unique presentation leading them to meet fewer ADHD diagnostic criteria [17–19]. Girls tend to be diagnosed much later than boys [18, 20]. This delay is not seen in adulthood, where the mean age at the time of diagnosis has been estimated at 32.7 years with no significant sex differences [21]. However, change is on the horizon and recent studies show that girls are increasingly being identified as having ADHD [i.e. 3, 16, 22]. This improvement is however not sufficient; a recent study demonstrated that females with ADHD are still not identified appropriately in the diagnostic process and continue to be less likely to receive pharmacological treatment unless their symptom severity is high [23].

This increased recognition of ADHD in females is partly due to increasing awareness of sex differences in ADHD presentations [e.g. 5]. Males with ADHD tend to show more core symptoms of the ADHD-I (inattentive), HI (hyperactive/impulsive) and C (combined) presentations, than their female counterparts [e.g. 24]. ADHD-I is more characteristic of females whereas ADHD-HI is more prevalent in males [5, 25, 26]. The HI presentation is associated with impulsive and hyperactive behaviours, while the I presentation is associated with lower levels of arousal, inattention and withdrawal [18]. Symptoms of ADHD-I, more characteristic of females, are often reflected in mood- or emotional dysregulation making differentiated diagnostics quite difficult [5], leading to misdiagnosis with internalizing disorders such as mood- or anxiety disorders, or depression [18, 21]. When a misdiagnosis is made, this in turn leads to inadequate and/or postponed treatment, fostering worse academic outcomes in the long run since basic skills are acquired in the elementary school years [17]. Apart from academic impairments, problems in psychosocial functioning in girls [3] and women [5] with ADHD have been reported repeatedly.

## Pharmacotherapy in ADHD

Pharmacotherapy is currently the first-line treatment in the management of ADHD symptoms [27–29]. Observational studies and clinical-trials, have shown that pharmacotherapy is effective in reducing core ADHD symptoms, improving daily functioning and may have a positive impact on self-esteem, both in the long and short-term, generally improving quality of life [30]. Currently, in the U.S., almost two thirds (62.0%) of children with ADHD aged two to 17 were taking medication for the treatment of ADHD symptoms [31]. Research on the safety, efficacy and effectiveness of ADHD medication in adults has grown in the past 15 years when it became known that ADHD often continues into adulthood and adversely affects functional outcomes. The difference between efficacy and effectiveness of medication lies in whether the focus is on the maximum possible effect of a certain medication, under ideal, standardized circumstances (efficacy) or on whether, and to what extent, the medication has a useful clinical effect for the patient (effectiveness) [i.e. 32, 33]. With respect to adverse effects of ADHD medication, findings are in general inconsistent. They vary from an increased sleep-onset latency, increased blood pressure, interference with growth (height and weight) to cardiac problems, increase in heart rate, and an association with tics and Tourette Syndrome [34].

With regard to ADHD pharmacotherapy, two classes of medication exist; stimulants and non-stimulants [35]. Stimulants include non-amphetamines (Methylphenidate (MPH), dex-methylphenidate (dexMPH)), amphetamines (dextroamphetamines (dexAMP), mixed amphetamine salts (MAS-XR)) and arousal agents [35]. MPH, an oral stimulant, is one of the most commonly used psychostimulants worldwide [36–38]. Like the other stimulants, it can be prescribed either as short-acting or as long-acting agent [35], depending on the type of impairment and impact on daily functioning. Behaviourally, studies have shown enhanced alertness as well as both reduced antisocial behaviours and impulsivity following stimulant drug treatment [39–41]. The effects of stimulants on cognition appear to be considerably smaller than the effects on behavioural symptoms related to ADHD [7], but MPH may positively affect several aspects of cognition, such as memory, self-regulation, certain domains of attention, executive functioning and visuospatial functioning have also been found to be positively affected by MPH [39, 40, 42–45]. Interestingly, lower doses of stimulants appear cognition-enhancing, while higher doses of stimulants are behaviourally activating but cognition-impairing [46, 47]. Due to the findings of positive effects of stimulants on behaviour and cognition, it has been assumed that consequently stimulants also have positive effects on academic performance [48]. However, although statistically significant improvements in academic performance are being reported in literature, in most longitudinal studies the–small- effect sizes of such improvements are such that any clinical relevance is doubtful [49, 50]. Children and adolescents with ADHD do not tend to perform at the same level as their peers following treatment, even if some improvement has occurred.

Even though stimulants are generally effective in most patients with ADHD, non-stimulant medication is also prescribed, because in approximately 25–30% of patients, ADHD symptoms are not adequately controlled by stimulant therapy, and not all individuals respond well to stimulant treatment [51, 52]. Atomoxetine (ATX) is the most commonly used non-stimulant, but Clonidine HCL extended-release and Guanfacine extended-release are also prescribed. Although several studies have revealed that the efficacy and effectiveness of ATX is comparable to that of MPH [e.g. 53–55], others have demonstrated that their efficacy and effectiveness may not be as robust as with stimulants, nor may they be superior in terms of tolerability [56].

## Sex and pharmacotherapy

Only in recent years the medical sciences have become increasingly aware of the impact of sex on the effects of pharmacotherapy in general. Female sex is increasingly seen as a risk factor for clinically relevant adverse drug reactions (ADRs) [57–59], with women having 1.5 to 1.7 times higher risk of developing an ADR to any type of medication for any kind of medical issue or illness [60]. Potential explanations for this phenomenon are sex-related differences in pharmacokinetic, immunological and hormonal factors and medication use exist, as well as the fact that women tend to have lower lean body mass [60]. This is illustrated by striking findings on sex-differences in the effects of heart medication. Santema and colleagues [61] conducted a study of 1308 men and 402 women from various countries, aimed at identifying optimal doses of heart failure medications. Women benefited from a significantly lower (50%) dose of the two main types of heart medication than is currently prescribed by the international cardiology guidelines. This was a worrying finding as women have been receiving much too high doses of heart medication accompanied by significant risks to their health. The authors propose that new medication should only be allowed to enter the market if a certain specified percentage of women was included in the sample, in order to determine the optimal female-specific dose [61]. Unfortunately, in the context of ADHD pharmacotherapy such research on identifying optimal doses for men and women is lacking.

## Study aims

As early as two decades ago, an important conclusion of the National Institute of Mental Health's Conference on Sex Differences in ADHD was that inferences drawn from studies in boys with ADHD cannot be applied routinely to girls with ADHD [62]. In recent years the difference in pharmacological response between females and males has received more attention. Therefore, this review aims to investigate sex differences in prescription rates, effectiveness and efficacy of pharmacotherapy treatment in girls and women with ADHD, and to identify gaps in the scientific knowledge on this topic.

## Method

### Literature search

This systematic literature review was performed conforming to the guidelines of Preferred Reporting Items for Systematic Reviews and Meta-Analyses (PRISMA) (see S3 Table for a checklist of the PRISMA guidelines for this study). Several procedures were used to identify potential studies for this review. First, the literature was searched electronically in PsychINFO, PubMed, and Web of Knowledge including all of the available literature up until the date of September 20th, 2019. The primary keywords 'ADHD' or 'attention deficit hyperactivity disorder' were combined with the secondary keywords 'pharmacotherapy', 'effectiveness', and 'efficacy'. These keywords were then combined with the sex-related keywords 'females', 'girls', 'women', 'gender' or 'sex', as well as with keywords relating to ADHD pharmacotherapy, such as 'methylphenidate', 'MPH', 'amphetamine', 'AMP', dextroamphetamine', 'dAMP', 'levoamphetamine', 'lAMP','lisdexamphetamine', 'lisdexAMP', 'atomoxetine', 'ATX', clonidine' or 'guanfacine'. In order to be included in this systematic literature review, studies had to meet the following inclusion criteria: (a) the study was written in English; (b) the ADHD sample was formally diagnosed with this disorder; (c) the study provided data explicitly sorted by sex; and (d) the study provided data on either prescription rates of ADHD pharmacotherapy or on the efficacy or effectiveness of stimulant or non-stimulant ADHD pharmacotherapy. Due to the scarce body of literature, no strict excluding criteria for the measurement of efficacy / effectiveness were set. The combination of the primary keywords with the keywords related to either sex or pharmacotherapy yielded an overall total of 2671 records. These initial results were then filtered to exclude duplicates and articles not fulfilling the inclusion criteria. Most articles were excluded for not explicitly reporting the data by sex. Following this procedure, eleven relevant articles remained. Subsequently, reference lists of these relevant articles were scanned for further potential articles. This search generated another three articles that fulfilled all set criteria and were therefore included. Hence, 14 articles were included in the review on the efficacy or effectiveness of stimulant or non-stimulant ADHD pharmacotherapy. The results were grouped by ADHD pharmacotherapy type; i.e. stimulants and non-stimulants, and age; i.e. children and adolescents (<18 years), and adults (≥ 18 years). Next, the same search method was conducted for studies on prescription rates, with the exception of the secondary keywords, which were 'prescription' or 'prescription rate'. After completing the search process, seven studies were found and the results were grouped by age; i.e. children and adolescents (<18 years), and adults (≥ 18 years). See Fig 1 for a flow chart of this process leading to a total of 21 included studies.

### Statistical description

Regarding the analysis of results on efficacy / effectiveness of ADHD pharmacotherapy, effect sizes of group differences were either derived from the original studies or calculated by hand

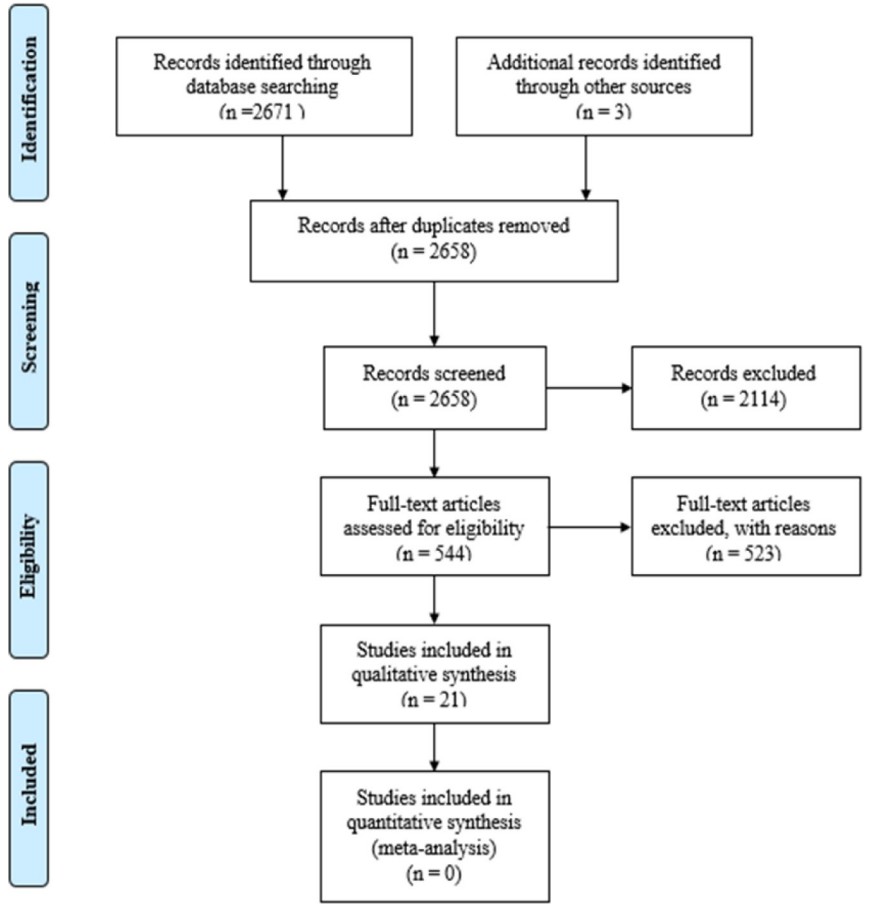

**Fig 1. Flow chart study inclusion process.**

by the first author in order to determine the size of between-sex differences. In some studies, efficacy or effectiveness of pharmacotherapy was investigated in all-female samples, by using a treatment group versus a placebo group. The effect size of choice was Cohen's $d$, which differentiates small ($d = 0.2$), medium ($d = 0.5$) and large ($d = 0.8$) effects [63]. In the studies on prescriptions rates of ADHD pharmacotherapy, percentages of prescription rates were mostly used hence the results of this study also uses percentages to compare sex-differences in prescription rates.

## Results

### Sex differences in prescription rates of ADHD pharmacotherapy

The results obtained in this review suggest that, overall, both girls and women with ADHD were significantly less likely to be prescribed pharmacotherapy than boys and men with this disorder, although the difference is less pronounced in adults [64–70]. Only one, population based, study demonstrated higher prescription rates in women with ADHD over 16 years of age, than in men with ADHD in that same age-group [68]. An overview of the seven reviewed studies is presented in S1 Table.

**Children and adolescents.** Four studies examined prescription rates in children and adolescents only, while two studies included both youngsters and adults in their study. Klein and

colleagues [65] examined MPH, dexAMP, mixed amphetamine salts (MAS), lisdexamphetamine, ATX, guanfacine and clonidine use in adolescents, and found that girls received one third of the prescriptions boys did (25.2% vs 74.8%). Similarly, a study of children and adolescents by Barbaresi and colleagues [64], showed that girls were significantly less likely to be prescribed MPH than boys (55.8% vs 69.7%; $d = 0.33$). Further, girls were more likely than boys to receive no medication at all (18.7% vs 28.4%; $d = 0.30$). No significant differences were however identified between boys and girls in the use of dexAMP, lAMP, Pemoline and methAMO. Although no significant sex-difference was identified in the use of dexAMP in the previous study; a large U.S. based study [66] consisting of 40,358 high-school students showed contrasting results. Girls were shown to receive significantly less prescriptions than boys for both dexAMP, ATX and MPH (2.1% vs 4.2% across medications). Girls under the age of 18 were also prescribed both ATX and MPH less frequently than boys in a Korean study by Song and Shin [67] covering the period 2007 to 2011. Specifically, MPH was prescribed to 70.3% (2007) and 67.4% (2011) of girls versus 74.8% and 71.1% respectively of boys. The same results were found for ATX (4.5% vs 6.1% in 2009; 10.7% vs 13.7% in 2011).

Two population-based studies including both children and adults replicated the above findings of significantly less prescriptions for girls under 18 compared to their male counterparts. Chang and colleagues [68] demonstrated that girls aged eight to 15 years received less pharmacotherapy, including MPH, AMP, dexAMP and ATX, than boys (34.9% vs 56.1%), while the opposite was found for older age groups (16–25 years: 32.1% of women vs 25.6% of men; 26–35 years: 16.2% of women vs 10.2% of men; 36–46 years: 16.8% of women vs 9.1 of men). Zoega and colleagues [69] also showed that girls aged seven to 15 received ADHD pharmacotherapy four times less than boys in this age group, with the opposite being the case for older age groups.

**Adults.** All studies including adult participants were population-based studies. One study examined prescription rates in adults only, while two studies included both adults and youngsters. A study on adults aged between 18 and 64, comparing use of ADHD medication across Nordic European countries [70] indicated that the number of women receiving medication, including MPH, ATX, AMP or dexAMP, was lower than those not receiving any medication (48% vs 52%). Looking at sex-differences; females consistently received less medication than males in all age groups. Interestingly this medication use declined over time for both women (from 0.93% of 18 to 24-year-olds to 0.18% of 45 to 64-year-olds) and men (from 1.19% of 18 to 24-year-olds to 0.21% of 45 to 64-year-olds). A similar study by Zoega and colleagues [69] on the use of MPH, AMP, dexAMP, Modafinil and ATX in adults and children showed that the sex difference visible in children was nearly gone in adults (21+ years of age). In contrast, the previously mentioned population-based study on both youngsters and adults aged between eight and 46-year-old, showed that adult women interestingly received *higher* rates of prescriptions than males. Specifically; among 16–25 year-olds, 32.1% of women vs 25.6% of men received medication. In the 26–35 year-old age group this was 16.2% vs 10.2% and among 36–46 year-olds it was 16.8% vs 9.1% [68].

## Sex differences in efficacy and effectiveness of ADHD pharmacotherapy

Sex differences in efficacy / effectiveness of different types of pharmacotherapy (stimulants and non-stimulants) are summarized for children / adolescents and adults respectively. In total, fourteen studies on the effects of ADHD medication on females and males with this disorder were included. An overview of the articles on efficacy / effectiveness of ADHD medication is presented in S2 Table.

**Stimulants.** *Methylphenidate–children / adolescents.* MPH efficacy / effectiveness differed between the sexes on some measures and was equal on others. One small (12 girls and 12 boys)

double-blind, placebo-controlled trial [71] examined several social, behavioural and academic variables in children aged under 12 with a diagnosis of ADHD-I, and found some sex differences. Compared to their male counterparts, when using MPH, girls showed less improvement on 'seatwork completion' (teacher-rating and observation) with an effect of ($d$ = 0.445). Teacher ratings also indicated that girls attempted less timed math questions ($d$ = 1.114) and scored lower on 'nonsense spelling' ($d$ = 1.919) than boys when taking MPH. However, when girls were off MPH, they showed a higher increase in conduct problems than boys ($d$ = 0.281). In contrast, girls with ADHD performed significantly better on focused attention than their male counterparts ($d$ = 0.96) on all doses of MPH in a study of 27 boys and 27 girls aged eight to 12, using a range of tests of attentional function [72]. Seventeen neuropsychological variables were measured, such as reduction of reaction time, errors on set-shifting, omission errors on sustained and divided attention. Although girls were slower than boys in tasks of sustained attention across all doses of MPH ($d$ = 0.58), they made less omission errors on all three doses ($d$ = 0.72) and scored higher on divided attention ($d$ = 0.76) on a low dose, compared to boys. A study by Wang, Cheng and Huang [73] on children and adolescents with ADHD treated with MPH, demonstrated that girls showed significantly lower symptom severity than boys after 24 months as per parent- and clinician ratings (b = 2.79, p = 0.018). However, teacher-ratings showed that only boys seemed to improve on symptom severity under the treatment of MPH; a result approaching significance (p = 0.058). Further, an attention test was used to examine neurocognitive functions and ADHD score. Attention scores significantly improved in boys using MPH (b = 0.07, p = 0.043), but not in girls, who performed poorer when using MPH at all moments (12, 18 and 24 months). In contrast, in a sample of children with ADHD aged up to 17 years (95 girls and 284 boys), the number of girls and boys with a positive response to MPH (73% vs 75%) and no response (17% vs 13%) did not differ significantly [64]. A recent study of female and male adolescents with ADHD aged between 13 and 18 (18 women and 32 men), comparing pre- and post-MPH treatment ratings of aspects of quality of life, found that MPH increased girls' self-rating of school functioning ($d$ = 0.60), but decreased their self-rating of physical functioning ($d$ = 0.54). In boys with ADHD, this lowered physical functioning was not seen; further, boys reported increased school- and psychosocial functioning ($d$ = 0.58 and $d$ = 0.40 respectively) following MPH use [74]. In addition, a small study using functional magnetic resonance imaging (fMRI) and functional connectivity analyses during a working memory task in five girls with ADHD aged 11 to 17 years [75] showed all girls were significantly more accurate when on MPH than when off medication ($d$ = 0.42).

A study by Sonuga-Barke and colleagues [76] on children with ADHD (48 girls and 136 boys) examined the efficacy of Concerta (CON), an extended-release MPH, Metadate controlled-delivery (MCD) and Equasyum XL (EQXL), an immediate-release MPH, on ADHD symptom severity. This study measured symptom severity ratings every 90 minutes over a twelve hour lab-school day, and showed that girls had significantly better scores than boys at 1.5 and 3 hours. At 4.5 and 6 hours girls and boys performed equally, but from 7.5 hours onwards the effect of MPH declined more quickly in girls. Children showed more improvement taking MCD-EQXL compared to CON after 90, 180 and 270 minutes, but the effects were smaller in girls (respectively $d$ = 1.09; $d$ = 1.19; $d$ = 1.10) compared to in boys (respectively $d$ = 2.32; $d$ = 2.39; $d$ = 1.66). At twelve hours, CON was superior to MCD-EQXL, but the effect was still smaller in girls ($d$ = 0.62) than in boys ($d$ = 1.24).

*Methylphenidate–adults*. One study examined the efficacy / effectiveness of MPH in adults (1,414,183 women and 1,579,704 men). Quinn and colleagues [77] showed a slightly lesser effect of ADHD medication in adult women compared to men. They investigated the risk of substance-related events (SREs) related to stimulant and non-stimulant use by means of an examination of US health care claims in the period 2005 to 2014. In this study, an SRE was

identified if (a) the patient received any substance use disorder diagnosis and (b) the service sub-category indicated that the encounter occurred in an emergency room or department [77]. Included stimulants were both MPH and several variations of AMP. The study demonstrated that during months in which women were medicated, their risk of an SRE was 31% lower compared to months in which they did not use ADHD medication. This was less than in men, who had a 35% lower risk of an SRE while on medication. Interestingly, two full years after ADHD medication was used, women still had a 14% lower risk of an SRE, again less than their male counterparts (19% lower).

*Mixed amphetamine salts–extended release (MAS-XR)–children and adolescents.* In 6–-12-year-old girls with ADHD using MAS-XR (N = 26) or ATX N = 31), one study [78] demonstrated a significantly increased effect of MAS-XR on both manners at school as well as attention compared to ATX, as rated by teachers on the Swanson, Kotkin, Agler, M-Flynn & Pelham (SKAMP) rating scale [79]. Girls taking MAS-XR tried to solve more items than girls taking ATX, but they did not differ on accuracy.

*Dextroamphetamine (dexAMP, Lisdexamphetamine (lisdexAMP) & Amphetamine (AMP)– children and adolescents.* In a sample of children and adolescents with ADHD [64], girls responded less favourably to dexAMP than boys (51% favourable response in females vs 78.8% in males). A lesser effect of dexAMP, lisdexAMP and AMP in females was also found in the large US study of healthcare claims discussed above, where females had a 31% lower risk of an SRE when on medication for ADHD, compared to males, whose risk was 35% lower when on medication. In contrast, a population-based study (12,503 girls and women and 26,249 boys and men) showed no sex difference (females $d = 0.21$ vs males $d = 0.22$) in the effect of AMP and dexAMP on depression with ADHD [68]; both sexes were less likely to suffer from depression than people who did not take medication.

**Non-stimulants.** *Atomoxitine (ATX)–children & adolescents.* In a pooled analysis of five studies [80] among children aged 6 to 15 years, (N = 136 girls and N = 658 boys) with the aim of evaluating treatment differences regarding health-related quality of life and ADHD symptoms across sexes, it was shown that girls with ADHD taking ATX showed a significantly lower tendency towards behaviours with adverse consequences in general ($d = 0.6$) and pertaining to themselves ($d = 0.6$) compared to the control placebo group. These effects were smaller in boys with ADHD ($d = 0.5$ and $d = 0.5$ respectively). Girls with ADHD taking ATX did not change on some subscales on which their male counterparts improved (i.e. self-content, $d = 0.2$; behaviours disturbing the process of growing up, $d = 0.4$; ability to solve interpersonal conflicts, $d = 0.2$; educational development relative to age, $d = 0.5$; skills in the classroom, $d = 0.4$; connections to classmates, $d = 0.4$; total, $d = 0.4$.).). Further, 7 to 13 year old girls taking ATX (N = 30) compared to placebo (N = 21) had significantly lower symptom severity on inattentive and hyperactive subscales, and on the total score in a study based on two randomized controlled trials [81]. Finally, 6 to12-year-old girls with ADHD using MAS-XR (N = 26) or ATX (N = 31) showed a significantly increased effect of MAS-XR on both manners at school as well as attention compared to ATX, as rated by teachers. Girls taking MAS-XR tried to solve more items than those taking ATX, but they did not differ on accuracy [81].

*Atomoxetine (ATX)—adults.* In a double-blind, placebo-controlled study (188 women and 348 men) investigating the effects of ATX on quality of life, emotional dysregulation and symptom severity, women consistently showed larger treatment effects than men [5]. However, this effect was only significant for two domains, emotional dysregulation and social life, in which women with ADHD improved significantly more than their male counterparts ($d = 0.28$ and $d = 0.19$ respectively), suggesting a higher efficacy of ATX. Further, although the total scores on an ADHD rating scales improved for both sexes; women with ADHD improved somewhat more than men with this condition ($d = 0.3$). Similarly, in a longitudinal study [82]

on adults with ADHD (137 women and 247 men), the scores on an adult-specifc attention deficit disorder scale increased significantly more in women with ADHD ($d$ = 1.7) than in men with ADHD ($d$ = 1.0). Women improved significantly more on the subscales hyperactivity and impulsivity ($d$ = 0.2), emotional dysregulation ($d$ = 0.3) and on the total score ($d$ = 0.3). On the CAARS, women with ADHD also improved significantly more compared to men with this disorder when using ATX ($d$ = 1.8 vs $d$ = 1.6).

**Adverse effects of ADHD pharmacotherapy.** Twelve of the fourteen studies included in this part of this review did not report on types of adverse effects of ADHD pharmacotherapy, while one of these studies did report on rates of adverse effects but not on type of such effects. The latter study, by Barbaresi and colleagues [64], compared boys and girls with ADHD who used medication and found no differences in the rate of occurrence of adverse effects. Although not specifying the type of adverse effects, Barbaresi and colleagues [64] moreover found that the occurrence of adverse effects was significantly greater for episodes of treatment with dexAMP (10.0%) than for MPH (6.1%) in both boys and girls. In the two all-girls studies by Biederman and team [78,81] adverse effects were examined and varied per study. While in the 2002 study [81] the most common adverse effects of ATX were rhinitis (25.8%), abdominal pain (29.0%) and headache (25.8%); the 2006 study [78] found somnolence (28.1%), upper abdominal pain (15.6%) and vomiting (15.6%) to be most common when using ATX. The 2006 study [78] also identified the most common side effects of the stimulant MAS-XR to be decreased appetite (40.7%), upper abdominal pain (29.6%) and insomnia (25.9%).

## Discussion

### Prescription rates

In children, girls under 18 years of age received significantly less ADHD medication than boys in all reviewed studies. In 13 out of 14 studies, adult women received less medication than adult men; this difference was however much less pronounced in adults compared to in children. One study [64] showed girls were much more likely than boys to be prescribed no pharmacotherapy at all, and one study found higher prescription rates in women with ADHD over 16 years of age than in men in the same age-group.

**Stimulant efficacy / effectiveness.** Although many studies found the overall improvement on symptom severity measures to be similar for females and males, several noteworthy sex differences were found. First, girls showed significantly lower symptom severity in the longer term (i.e. after 24 months) than boys by parent- and clinician' ratings [73], but not by teacher-rating. This discrepancy between parent/ clinician's ratings and teacher's ratings suggests that the effect of MPH may vary in different settings. A decrease in the often disruptive behaviours of boys with ADHD, especially those with the HI presentation, may have been less obvious in family life than the rather withdrawn behaviour of girls. In the classroom setting however, boys' improvements following MPH pharmacotherapy may have been more visible to teachers than the changes that occurred in girls. Support for this assumption comes from the finding of sex differences in certain domains of attention, with boys with ADHD slightly outperforming girls with ADHD on tasks of focused and sustained attention when using MPH [72]. Similarly, improvements in inattention and hyperactivity/impulsivity were less in females with ADHD than in their male counterparts when using MPH, according to teacher-ratings [73]. Such sex differences in attentional performance are interesting in light of presentations of ADHD, as females are most often diagnosed with the inattentive presentation. Considering girls predominantly display attentional deficits and less pronounced hyperactivity and impulsivity, further research on whether MPH is a suitable pharmacotherapy for girls is imperative. Further, one study [76] found that girls with ADHD had a stronger effect of MPH immediate-release earlier

in the day but also experienced an earlier decline compared to boys with this disorder. These findings were in line with medical research that showed that while females reported to experience the effect of stimulants earlier than males, bioavailability of orally administered MPH was found to be significantly lower for females than for males [83]. Such biological sex-differences could explain the finding of an earlier decline in MPH efficacy in girls. A once-daily application of MPH may therefore not be optimal for girls.

Research on sex-differences in efficacy of dexAMP is scarce; the two studies [64, 68] examining this type of stimulant medication showed either no sex-difference or less favourable treatment outcomes in girls and women compared to boys and men. One study found lower rates of depression in both females and males when using dexAMP. These findings suggest that more research is needed to further examine the use of dexAMP in the context of female ADHD.

**Non-stimulant efficacy / effectiveness.** When using ATX, girls and women with ADHD were found to improve more on symptom severity, hyperactivity, impulsivity and emotional dysregulations/emotional factors than their male counterparts. They also demonstrated lower symptom severity scores of core ADHD symptoms than males in the studies in this review. Similar findings were demonstrated when the comparison was made between females on ATX and those using a placebo. These positive effects of ATX are promising first results that require replication.

**Sex-differences in pharmacokinetics.** Although studies specifically focusing on sex differences in efficacy or effectiveness of ADHD pharmacotherapy are scarce, recent studies show females may respond differently than males [58,59]. The dosage has to be adjusted to body mass index and body composition, the latter of which is known to differ across the sexes [83]. Pharmacokinetics is also influenced by sex; the knowledge that differences in drug metabolism can lead to different efficacy or effectiveness outcomes is well-established in the medical field [83]. Sex-differences in pharmacokinetics have been able to explain some of the findings of the current review. One explanation for the findings of less favourable outcomes in girls and women using dexAMP compared to their male counterparts could be the influence of hormones, in particular as one of the samples included adolescents. Levels of estrogen and progesterone fluctuate among the menstrual cycle and differently influence the effect of stimulant drugs at different points of the month in adolescent and adult females [84] Unfortunately, it is unclear from the data provided by the population-based study why the outcomes were rated as not favourable. It is possible that unstable drug effects with treatment effectiveness depending on the phase of the menstrual cycle were reported as unfavourable by female patients. After all, evidence exists that amphetamines in particular, unlike other substances, interact markedly with female sex hormones [85]. Further, pharmacokinetic research showing that while females reported to experience the effect of stimulants earlier than males, bioavailability of orally administered MPH was found to be significantly lower for females than for males [86]. Such biological sex-differences could explain the finding of an earlier decline in MPH efficacy in girls. Future studies should investigate differing dosage patterns of stimulants adapted to the fluctuating hormone levels in the course of a menstrual cycle, as fluctuating hormones were found to influence cognitive functioning differently at different phases of the female menstrual cycle also in the absence of drug treatment [87]. Overall, the long-term response to ADHD pharmacotherapy may differ between sexes due to genetic variability [88]. Determining sex differences in the longitudinal course of ADHD would positively impact on clinicians' prognoses as well as the provision of adequate intervention for patients.

**Adverse effects.** Sex differences in adverse effects have not been systematically investigated in the reviewed studies. Only two studies examined adverse effects of female-specific pharmacotherapy and results were contradictory. Thus, with such limited findings, no conclusions can be drawn.

**Suggestions for future research on female ADHD pharmacotherapy.** This review provides evidence that different pharmacological approaches to the treatment of ADHD have their own unique effect on females and males, which should be further investigated. Females are most often diagnosed with the inattentive presentation of ADHD [5], hence considering attentional deficits when prescribing ADHD pharmacotherapy may be beneficial. This review found that stimulant drugs may not be the ideal treatment of attentional deficits for females with ADHD, due to inconsistent results and potentially different pharmacokinetics, which needs to be examined further. As research on female-specific ADHD in general and female ADHD pharmacotherapy more specifically is scarce to the extent that only 11 studies could be included in this review, this again highlights that much more research needs to be conducted in this field. However, the scarcity of such research is not surprising; it is an observation often seen in medical research [89]. It would be helpful if authors reporting the results of clinical trials would ensure that their data is analysed and reported separately for female and male participants [89]. It is paramount to do so because incorrect conclusions may be drawn when female and male results are combined, camouflaging differences between sexes. Second, reporting data by sex facilitates meta-analyses, and these analyses are very relevant in order to draw sex-specific conclusions over large sample sizes. In the present review this was not possible due to heterogeneity between studies and insufficient reporting of statistical outcomes. Not only would it be helpful if findings were reported by sex, this should also be done in a standardised manner in order to shed light on how they function independently and together and have their impact on health care. Finally, unnecessary research should be avoided; repeating a trial because previous relevant studies did not report results by sex can be considered unethical [89]. Biological sex is to date only recognized implicitly as an important factor in clinical research. In order to improve understanding and comparability across studies, it has been suggested with respect to reporting data that the term sex should be reserved for reporting biological factors, while 'gender' should be used when reporting gender identity or psychosocial or cultural factors [89].

Lastly, the lack of information on pharmacokinetics as well as on adverse effects of ADHD pharmacotherapy in the studies encompassing the present review is an issue that should be addressed. Although it was a topic beyond the scope of this review and hence a separate search on this topic was not carried out, it is recommended that future studies examine sex differences in adverse effects of ADHD medication as well as pharmacokinetics.

**Strengths and limitations.** As with all reviews, the current study has some limitations. The finding that ATX may be a promising medication for girls and women with ADHD is based on a limited number of studies, hence the robustness of these results should be further investigated. The rate of female participants, inclusion of ADHD presentations, investigated domains and outcome measures varied between studies. Study designs differed as well; ranging from setting, such as classrooms [70, 74, 77], in—and outpatient centers [5,71,80,81], to experimental laboratory [72]. While this may be regarded a strength, as it considers the multi-factorial approach of studying ADHD and treatment outcome, the diversity in designs and methods were a limitation in the analyses of this review as it increased heterogeneity between studies which obstructed comparison between studies and integration of findings. Further, we may assume that the prescription of medication to ADHD patients was generally based on clinicians' discretion and patients'/caregivers' willingness in the clinical settings. Therefore, the effects are likely to have been influenced by selection bias or rating bias. Some studies used pooled analysis [80] while others used population-based study [64], and the type of pharmacotherapy use also differed across studies, making a direct comparison of outcomes difficult. Lastly, some reviewed studies had a small sample size. As is known, small sample sizes produce less reliable results and often conclusions cannot, or merely tentatively be replicated.

## Conclusion

This review leads to several main conclusions. First and foremost, due to its great impact on the ability to draw significant conclusions, only very few of the many ADHD medication trials published explicitly sorted their medication trials by sex (k = 14). Further, sex differences in adverse effects of ADHD pharmacotherapy have hardly been investigated (k = 1), although prescription rates are rapidly rising. Despite the overall rise in prescription rates, girls with ADHD are consistently prescribed less ADHD pharmacotherapy than boys with this disorder. In the adult ADHD population however, this gap is narrowing and adults are approaching similar rates of prescription. Finally, each pharmacotherapy for ADHD has unique effects in males and females with ADHD; hence some sex differences were identified: a) lower symptom severity (i.e. more improvement) in the longer term in girls with ADHD using MPH, not the case in their male counterparts, b) less improvement in inattention, hyperactivity and impulsivity in girls and women with ADHD using MPH than in the male ADHD population, c) in girls with ADHD a stronger effect of MPH earlier in the day but also an earlier decline compared to boys with ADHD–single daily use of MPH possibly not be optimal for girls, d) effectiveness / efficacy of non-stimulant medication was stronger for girls and women with ADHD compared to boys and men with ADHD on symptom severity, hyperactivity, impulsivity and emotional dysregulation / emotional factors. ATX may be a promising medication for girls and women with ADHD.

The authors call for more research on female-specific pharmacotherapy that is sorted by sex, a suggestion that can also be accomplished by re-analysing existing medication trials. Further suggestions on reporting sex-specific data are also made.

## Supporting information

**S1 Table. Sex differences in ADHD pharmacotherapy prescription rates.**
(DOCX)

**S2 Table. Summary of included studies on effects of pharmacotherapy on females and males with ADHD.**
(DOCX)

**S3 Table. PRISMA 2009 checklist.**
(DOCX)

## Author Contributions

**Conceptualization:** Francien M. Kok.

**Data curation:** Francien M. Kok.

**Formal analysis:** Francien M. Kok.

**Funding acquisition:** Francien M. Kok.

**Investigation:** Francien M. Kok.

**Methodology:** Francien M. Kok.

**Project administration:** Francien M. Kok.

**Resources:** Francien M. Kok.

**Software:** Francien M. Kok.

**Supervision:** Francien M. Kok, Yvonne Groen.

**Validation:** Francien M. Kok.

**Visualization:** Francien M. Kok.

**Writing – original draft:** Francien M. Kok.

**Writing – review & editing:** Francien M. Kok, Yvonne Groen, Anselm B. M. Fuermaier, Oliver Tucha.

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
