## [Decision Letter · Decision Letter 0]

15 Jun 2020

PONE-D-19-29154

The female side of pharmacotherapy for ADHD – a systematic literature review.

PLOS ONE

Dear Dr. Kok

Thank you for submitting your manuscript to PLOS ONE. After careful consideration, we feel that it has merit but does not fully meet PLOS ONE’s publication criteria as it currently stands. Therefore, we invite you to submit a revised version of the manuscript that addresses the points raised during the review process. 

Kindly review the manuscript addressing the issues raised by the reviewers. Please also ensure that the typographical errors are corrected.

We look forward to receiving your revised manuscript.

Kind regards,

Gerard Hutchinson, MD

Academic Editor

PLOS ONE

Journal Requirements:

2. Please include a copy of Table 1 which you refer to in your text on page 12 and Table 2 onb page 14.

3. Please upload a copy of Supporting Information Table S3 which you refer to in your text on page 33.

Reviewers' comments:

Reviewer's Responses to Questions

**Comments to the Author**

1. Is the manuscript technically sound, and do the data support the conclusions?

Reviewer #1: Yes

Reviewer #2: Partly

2. Has the statistical analysis been performed appropriately and rigorously? 

Reviewer #1: Yes

Reviewer #2: N/A

3. Have the authors made all data underlying the findings in their manuscript fully available?

Reviewer #1: Yes

Reviewer #2: Yes

4. Is the manuscript presented in an intelligible fashion and written in standard English?

Reviewer #1: Yes

Reviewer #2: Yes

5. Review Comments to the Author

Reviewer #1: This comprehensive review examined sex differences in prescription rates and efficacy or effectiveness of pharmacotherapy in girls and women with ADHD. The authors found sex-differences are demonstrated in the prescription, usage and efficacy/effectiveness of both stimulant and non-stimulant ADHD pharmacotherapy. The authors suggest that a single daily use of MPH may possibly not be optimal for girls with ADHD and ATX may be a promising medication for girls and women with ADHD. This paper is generally well-written and has several merits. There are some minor issues to be addressed:

1. The authors suggest that a single daily use of MPH may possibly not be optimal for girls with ADHD and ATX may be a promising medication for girls and women with ADHD. I think this statement is somewhat over-concluded. There are only 5 studies about ATX recruited in this study. The robustness of the results should be further investigated.

2. The recruitment criteria were not limited to randomized, controlled, double-blinded trials. Because the prescription for ADHD patients were generally based on clinicians’ discretion and patients’/caregivers’ willingness in the clinical settings, the effect were likely influenced by selection bias or rating bias.

3. If the authors considered that observational studies, not only randomized controlled trials, are also important for assessing efficacy or effectiveness of pharmacotherapy. I suggest some epidemiological studies using claims data should also be considered in this study, and the indicators of effectiveness may be set as discontinuation rates or co-occurrence of comorbidity.

Reviewer #2: This paper intends to examine an interesting topic summarizing differences between males and females with ADHD, regarding their clinical presentation, and response to pharmacotherapy, based on a literature review. The authors appropriately suggest two main aims, which include reporting on differences between females when compared to males regarding rates of prescriptions of ADHD meds, and responses to such medications regarding effectiveness or efficacy. The authors also review the literature in order to compare the sexes based on their tolerability to medications. I liked the topic, but found the paper hard to read due to its typographical and other errors. Please see the attached review.

6. PLOS authors have the option to publish the peer review history of their article (what does this mean?). If published, this will include your full peer review and any attached files.

Reviewer #1: No

Reviewer #2: No

---

## [Author Response · Author response to Decision Letter 0]

6 Jul 2020

Please see attached Rebuttal letters.

---

## [Decision Letter · Decision Letter 1]

3 Sep 2020

The female side of pharmacotherapy for ADHD – a systematic literature review.

PONE-D-19-29154R1

Dear Dr. Kok,

We’re pleased to inform you that your manuscript has been judged scientifically suitable for publication and will be formally accepted for publication once it meets all outstanding technical requirements.

Kind regards,

Gerard Hutchinson, MD

Academic Editor

PLOS ONE

Additional Editor Comments (optional):

Reviewers' comments:

Reviewer's Responses to Questions

**Comments to the Author**

1. If the authors have adequately addressed your comments raised in a previous round of review and you feel that this manuscript is now acceptable for publication, you may indicate that here to bypass the “Comments to the Author” section, enter your conflict of interest statement in the “Confidential to Editor” section, and submit your "Accept" recommendation.

Reviewer #1: All comments have been addressed

2. Is the manuscript technically sound, and do the data support the conclusions?

Reviewer #1: Yes

3. Has the statistical analysis been performed appropriately and rigorously? 

Reviewer #1: Yes

4. Have the authors made all data underlying the findings in their manuscript fully available?

Reviewer #1: Yes

5. Is the manuscript presented in an intelligible fashion and written in standard English?

Reviewer #1: Yes

6. Review Comments to the Author

Reviewer #1: The authors have adequately address my questions. The authors make a great contribution for this research field. I have no further comment.

7. PLOS authors have the option to publish the peer review history of their article (what does this mean?). If published, this will include your full peer review and any attached files.

Reviewer #1: No

---

## [Editor Report · Acceptance letter]

10 Sep 2020

PONE-D-19-29154R1 

The female side of pharmacotherapy for ADHD – a systematic literature review 

Dear Dr. Kok:

I'm pleased to inform you that your manuscript has been deemed suitable for publication in PLOS ONE. Congratulations! Your manuscript is now with our production department. 

Kind regards, 

on behalf of

Dr. Gerard Hutchinson 

Academic Editor

PLOS ONE